# Nisin, a Probiotic Bacteriocin, Modulates the Inflammatory and Microbiome Changes in Female Reproductive Organs Mediated by Polymicrobial Periodontal Infection

**DOI:** 10.3390/microorganisms12081647

**Published:** 2024-08-12

**Authors:** Changchang Ye, Chuanjiang Zhao, Ryutaro Kuraji, Li Gao, Hélène Rangé, Pachiyappan Kamarajan, Allan Radaic, Yvonne L. Kapila

**Affiliations:** 1Orofacial Sciences Department, School of Dentistry, University of California, San Francisco, CA 94143, USA; yechangchang@scu.edu.cn (C.Y.); zhaochj@mail.sysu.edu.cn (C.Z.); r-kuraji@tky.ndu.ac.jp (R.K.); gaol7@mail.sysu.edu.cn (L.G.); helene.range@aphp.fr (H.R.); pkamarajan@dentistry.ucla.edu (P.K.); aradaic@dentistry.ucla.edu (A.R.); 2State Key Laboratory of Oral Diseases, National Clinical Research Center for Oral Diseases, Department of Periodontology, West China Hospital of Stomatology, Sichuan University, Chengdu 610041, China; 3Department of Periodontology, Guanghua School of Stomatology, Hospital of Stomatology, Sun Yat-sen University, Guangzhou 510275, China; 4Department of Periodontology, The Nippon Dental University School of Life Dentistry at Tokyo, Tokyo 102-8159, Japan; 5Department of Periodontology, UFR of Odontology, University of Rennes, 35000 Rennes, France; 6Service d’Odontologie, CHU de Rennes, 35000 Rennes, France; 7Sections of Biosystems and Function and Periodontics, School of Dentistry, University of California, Los Angeles, CA 90095, USA

**Keywords:** oral microbiome, reproductive tract microbiome, periodontal disease, nisin, antimicrobial therapy

## Abstract

Periodontitis-related oral microbial dysbiosis is thought to contribute to adverse pregnancy outcomes (APOs), infertility, and female reproductive inflammation. Since probiotics can modulate periodontitis and oral microbiome dysbiosis, this study examined the effects of a probiotic bacteriocin, nisin, in modulating the reproductive microbiome and inflammation triggered by periodontitis. A total of 24 eight-week-old BALB/cByJ female mice were randomly divided into four treatment groups (control, infection, nisin, and infection+nisin group), with 6 mice per group. A polymicrobial (*Porphyromonas gingivalis*, *Treponema denticola*, *Tannerella forsythia*, *Fusobacterium nucleatum*) mouse model of periodontal disease was used to evaluate the effects of this disease on the female reproductive system, with a focus on the microbiome, local inflammation, and nisin’s therapeutic potential in this context. Moreover, 16s RNA sequencing was used to evaluate the changes in the microbiome and RT-PCR was used to evaluate the changes in inflammatory cytokines. Periodontal pathogen DNA was detected in the reproductive organs, and in the heart and aorta at the end of the experimental period, and the DNA was especially elevated in the oral cavity in the infection group. Compared to the control groups, only *P. gingivalis* was significantly higher in the oral cavity and uterus of the infection groups, and *T. forsythia* and *F. nucleatum* were significantly higher in the oral cavity of the infection groups. The infection and nisin treatment group had significantly lower levels of *P. gingivalis*, *T. forsythia*, and *F. nucleatum* in the oral cavity compared with the infection group. Since periodontal pathogen DNA was also detected in the heart and aorta, this suggests potential circulatory system transmission. The polymicrobial infection generally decreased the microbiome diversity in the uterus, which was abrogated by nisin treatment. The polymicrobial infection groups, compared to the control groups, generally had lower *Firmicutes* and higher *Bacteroidota* in all the reproductive organs, with similar trends revealed in the heart. However, the nisin treatment group and the infection and nisin group, compared to the control or infection groups, generally had higher *Proteobacteria* and lower *Firmicutes* and *Bacteroidota* in the reproductive organs and the heart. Nisin treatment also altered the microbiome community structure in the reproductive tract to a new state that did not mirror the controls. Periodontal disease, compared to the controls, triggered an increase in inflammatory cytokines (IL-6, TNF-α) in the uterus and oral cavity, which was abrogated by nisin treatment. Polymicrobial periodontal disease alters the reproductive tract’s microbial profile, microbiome, and inflammatory status. Nisin modulates the microbial profile and microbiome of the reproductive tract and mitigates the elevated uterine inflammatory cytokines triggered by periodontal disease.

## 1. Introduction

Pelvic inflammation is defined as a polymicrobial infection of the upper reproductive/genital tract (including the uterus, fallopian tubes, and ovaries) caused by infection with pathogens such as *Chlamydia trachomatis*, *Neisseria gonorrhoeae*, and *Mycoplasma genitalium* and is a leading cause of infertility in women [1]. Upper genital tract microbiome dysbiosis in women with unexplained infertility is of concern as it may play a role in the etiology of infertility.

Previously, the upper reproductive tract has been assumed to be mostly sterile, with the presence of bacteria indicating a probable disease state. Current knowledge, based on both culture and culture-independent data, has shown that the upper reproductive tract harbors its own microbiota, although with both a lower abundance and diversity than in the vagina [1,2]. Colonization of the female upper reproductive tract by microbes was generally not considered to induce a significant inflammatory response, indicating that the microbial community was benign [3]. However, microbiome dysbiosis of the upper reproductive tract might influence reproductive functions in health and disease.

An ascending route of infection from the lower genital tract, such as bacterial vaginitis, was believed to be the main pathogenetic mechanism [4]. Another possible route of upper reproductive tract inflammation might be through the circulation and as a result of oral pathogens. The effect of oral diseases, especially periodontal disease, on female fertility has been evaluated in recent studies. Cassini et al. [5] reported that periodontal pathogens were detected in human urogenital tract microflora, and the most representative species in the genital tract of the preterm group were *T. denticola*, *T. forsythia*, and *P. intermedia*. The presence of the periodontal pathogen *T. denticola* in the vaginal flora, regardless of the amount, was adversely associated with preterm delivery. Chen et al. [6] reported that dysbiosis of the reproductive tract microbiome was related to infertility, and specific bacteria in the uterus, such as *Actinomyces*, *Corynebacterium*, *Enterococcus*, *E. coli*, *Fusobacterium*, *Gardnerella*, *Prevotella*, *Propionibacterium*, *Staphylococcus*, and *Streptococcus*, have been associated with endometriosis, a pelvic inflammatory disease. McKinnon [7] reported a case of pelvic inflammatory disease associated with *F. nucleatum*. Therefore, these data support another possible pathogenic route for pelvic inflammation; namely, originating from the patient’s oral cavity.

In clinical studies, periodontal pathogens, especially *P. gingivalis* and *F. nucleatum*, have been widely detected in fetal–maternal organs, specifically the utero-placental tissues, such as the placenta, amniotic fluid, and umbilical cord blood [8,9,10]. Animal studies also demonstrated that oral infection with *P. gingivalis* or *F. nucleatum* leads to colonization in the mouse placenta, causing localized infection and increased levels of the pro-inflammatory cytokines interleukin-1, interleukin-6, interleukin-8, and tumor necrosis factor-alpha, leading to preterm and stillbirth [11,12,13]. These data indicate that periodontal pathogens are involved in the etiology and progression of placental inflammation.

Nisin, a type of class I antibiotic bacteriocin produced by *Lactococcus lactis*, has shown efficacy in treating a variety of infectious diseases, including gastrointestinal infections, respiratory tract infections, skin and soft tissue infections, mastitis, and other oral infectious diseases, including periodontal disease using in vitro and in vivo models [14,15,16,17,18,19,20,21,22]. Our previous studies revealed that nisin-producing *L. lactis* and nisin itself decrease the levels of periodontal pathogens in vitro and in vivo [15,16,17,18,19,20,21,22]. We further found that even low concentrations of nisin (0.5 mg/mL) can inhibit the growth of dental plaque biofilms in vitro [15,16,17,21,22]. In addition, in animal models, nisin abrogates periodontal pathogen infections and periodontal microbiome dysbiosis while promoting a shift in the oral microbiota toward a healthy state without causing damage to healthy oral cells, corroborating our in vitro data on oral cells [16,18]. Nisin can also prevent periodontal disease-related bone loss and inflammation while promoting reparative periodontal proliferation [18]. Furthermore, nisin also mitigates periodontal disease-mediated dysbiosis and inflammatory effects on the gut, liver, and brain [19,20].

Moreover, nisin can be used to treat genital tract infections. Recent clinical studies showed that *L. lactis*, the bacteria which produces nisin, has been detected as a symbiotic bacterium in the vagina of healthy women [23]. Furthermore, nisin can prevent sexually transmitted diseases [23]. Nisin can also treat bacterial and fungal vaginitis and down-regulate the expression of cytokines at this site [24].

In light of these combined data, evaluating the effects of periodontal pathogens on the female reproductive system microbiome is warranted. Since probiotics and probiotic bacteriocins can modulate oral and genital tract infections and microbiome dysbiosis, the aim of this study was to evaluate the influence of a polymicrobial periodontal infection on the reproductive organs in mice and the effects of a probiotic/lantibiotic bacteriocin, nisin, in modulating the reproductive microbiome and inflammatory changes triggered by periodontitis.

## 2. Material and Methods

### 2.1. Ethics Approval

The experimental procedures were performed in accordance with the guidelines of the Institutional Animal Care and Use Committee of the University of California, San Francisco (IACUC approval number: AN171564-01B).

### 2.2. Preparation of Periodontal Bacteria

In this study, *Porphyromonas gingivalis* (FDC 381), *Treponema denticola* (ATCC 35405), *Tannerella forsythia* (ATCC 43037), and *Fusobacterium nucleatum* (ATCC 10953) were cultured anaerobically (85% N_2_, 10% H_2_, 5% CO_2_) at 37 °C in an anaerobic chamber, as previously described [18,19,20,23]. For the oral polymicrobial infection, all 4 periopathogens were mixed in equal volume containing a total of 10^9^ CFU/mL of combined bacteria (2.5 × 10^8^ CFU/mL of *P. gingivalis*, 2.5 × 10^8^ CFU/mL of *T. denticola*, 2.5 × 10^8^ CFU/mL of *T. forsythia*, and 2.5 × 10^8^ CFU/mL of *F. nucleatum*) and allowed to interact. After 5 min, the bacterial consortia was mixed thoroughly with an equal volume of sterile 4% (*w*/*v*) carboxymethyl cellulose (CMC; Sigma-Aldrich, St. Louis, MO, USA) in PBS, and this mixture was used for the oral lavage [25].

### 2.3. Nisin Preparation

An ultra-pure (>95%) food-grade form of nisin Z (NisinZ^®^ P) was purchased from Handary (Evere, Belgium). The nisin stock solution was prepared as previously described [18,19,20]. Briefly, nisin powder was added to sterile Milli-Q water at a concentration of 600 μg/mL and mixed by rotation for 4 h. Next, the solution was filtered using a Millex 0.22 μm syringe filter (Sigma-Millipore, St. Louis, MO, USA) and stored at 4 °C until further use. The solution was stored for a maximum of 5 days before use. For oral treatment of the mice, 4% CMC was added to the nisin solution (at equal volumes), adjusting it to the final concentration of 300 μg/mL.

### 2.4. Infection of Mice

A total of 24 eight-week-old BALB/cByJ female mice (The Jackson Laboratories, Farmington, CT, USA) were housed in microisolator plastic cages and randomly distributed into 4 groups (N = 6 per group), including a control (CTL) group, infection group, nisin group, and infection+nisin (INF+nisin) group. A description of the infection protocols is shown in Figure 1. Prior to the polymicrobial inoculation, all the mice were given trimethoprim (0.17 mg per mL) and sulfamethoxazole (0.87 mg per mL) daily for 7 days in the drinking water and their oral cavity was rinsed with 0.12% chlorhexidine gluconate (Peridex) mouth rinse to inhibit the native oral microbiota [25,26].

The polymicrobial inoculum was administered topically in the morning for 4 consecutive days, every week, for a total of 8 weeks, via oral lavage. This model used 4 consecutive days of infection, thus enabling research staff to inoculate during a standard working week schedule [27,28,29]. A sterile 4% CMC solution was administered as the control treatment. Then, nisin (300 µg/mL, 0.2 mL per mouse) was administered every day in the evening, every week, for a total of 8 weeks, via oral gavage. Oral swab samples were collected at baseline and at the end of the experimental period (8th week of infection) to evaluate the microbial status. The samples were collected from the oral cavity using a sterile micro-sized cotton swab. Then, the cotton tips were immersed in 10:1 Tris-EDTA buffer immediately and stored at −80 °C until further processing for DNA isolation. At the end of 8 weeks, the mice were euthanized and oral swabs and uterus, ovary, vagina, heart and aorta tissues from each mouse were collected using sterile surgical methods. For DNA and RNA stabilization, the collected tissues were kept in RNAlater solution (Thermo-Fisher, Waltham, MA, USA) immediately after sample collection at −80 °C until further processing.

### 2.5. DNA Isolation from Samples/Tissues (Oral Swabs, Uterus, Ovary, Vagina, Heart and Aorta) to Evaluate Presence of Periodontal Pathogens via Reverse Transcription–Quantitative Polymerase Chain Reaction (RT-qPCR)

DNA isolated from the oral swabs and uterus, ovary, vagina, heart and aorta samples was used to evaluate the presence of the periodontal pathogens in the mice using methods described in our previous study [18]. For the oral swabs, DNA was isolated from the storage solution and purified using the QIAamp^®^ DNA Mini Kit (Qiagen, Venlo, the Netherlands). For the tissue samples, the collected tissue (≤10 mg) was powdered with a mortar and pestle under continuous liquid nitrogen, then DNA was isolated from the samples (uterus, ovary, vagina, heart and aorta) using the QIAamp^®^ DNA Mini Kit (Qiagen, Venlo, the Netherlands). A QuantStudio 3 qPCR system (Applied Biosystems, San Francisco, CA, USA) was used to quantify the periodontal pathogens from 100 ng of total DNA in the oral swabs and uterus, ovary, vagina, heart and aorta samples. Serial dilutions of known concentrations of the periopathogen DNA (ATCC, Manassas, VA, USA) were also analyzed via qPCR and used to prepare standard curves for quantification of the pathogens.

### 2.6. RNA Isolation from Samples/Tissues (Oral Swabs, Uterus, Ovary, Vagina, Heart, and Aorta) to Evaluate Immune Cytokine Levels via RT-qPCR

RNA isolated from the oral swabs, uterus, ovary, vagina, heart and aorta was used to evaluate the immune cytokine levels. Then, the tissue samples (≤10 mg) were powdered with a mortar and pestle under continuous liquid nitrogen, and RNA from the tissue samples was extracted using the RNeasy Mini Kit (Qiagen, Venlo, The Netherlands) according to the manufacturer’s manual. The RNA quality and quantity were assessed using a NanoDrop One (Thermo-Fisher, Waltham, MA, USA). Subsequently, 100 ng of total RNA was synthesized into cDNA using the SuperScript IV VILO cDNA Synthesis Kit (Thermo-Fisher, Waltham, MA, USA), according to the manufacturer’s manual, using a MyCyclar Thermo Cyclar (Bio-Rad, Hercules, CA, USA). Finally, the expressions of interleukin-1 beta (IL-1β), interleukin-6 (IL-6) and tumor necrosis factor-alpha (TNF-α) were analyzed using a QuantStudio 3 qPCR (Applied-Biosystems, San Francisco, CA, USA) and normalized to the expressed GAPDH in the samples.

### 2.7. 16S Sequencing of the DNA Isolated from Tissues (Uterus, Ovary, Vagina and Heart)

DNA from the uterus, ovary, vagina and heart (extracted as in Section 2.5. of this manuscript) was submitted to Novogene, Inc. (Sacramento, CA, USA) for 16S sequencing. Given the limited amounts of DNA harvested from the aorta, 16S sequencing of this tissue was not possible after the quantification of the periodontal pathogens and cytokine assay.

PCR amplification of the V3-V4 variable regions (341F-806R) was performed. PCR products with a proper size were then selected using 2% agarose gel electrophoresis. Equal amounts of PCR products from each sample were pooled, end-repaired, A-tailed and further ligated with Illumina adapters. Then, the libraries were sequenced on a paired-end NovaSeq 6000 System platform (Illumina, San Diego, CA, USA) to generate 250 bp paired-end raw reads. These paired-end reads were then truncated by cutting off the barcode and primer sequences and merged using FLASH (V1.2.7) [30]. Quality filtering on the raw tags was performed under specific filtering conditions to obtain the high-quality clean tags according to QIIME (V1.7.0) [31,32]. The tags were compared with the SILVA138 reference database using the UCHIME algorithm [33] to detect and remove chimera sequences, obtaining the effective tags.

Analysis of the obtained sequences was performed by Uparse software (v7.0.1090) [34]. Sequences with ≥97% similarity were assigned to the same OTUs, while representative sequences for each OTU were screened for further annotation. For each representative sequence, the QIIME (Version 1.7.0) in Mothur method was performed against the SSUrRNA database of the SILVA138 database to obtain species annotation at each taxonomic rank (threshold: 0.8~1) down to the species level [35,36,37]. To obtain the phylogenetic relationship of all the OTUs, representative sequences were obtained using MUSCLE (Version 3.8.31) [38]. The OTU abundance information was normalized using a standard of the sequence number corresponding to the sample with the least sequences. Subsequent analysis of the alpha diversity and beta diversity were all performed based on this normalized data.

### 2.8. Alpha Diversity

Four indices were used to evaluate the alpha diversity in the tissues, namely the observed species and Shannon, Simpson and Chao1 diversity indices. All the indices were calculated using the QIIME software (Version 1.7.0) and the graphs were generated using R software (Version 2.15.3) [39].

### 2.9. Beta Diversity

In this study, the beta diversity was measured via unweighted UniFrac with the QIIME software (version 1.7.0) [40]. Cluster analysis was preceded by principal component analysis (PCA), which was applied to reduce the dimensions of the original variables using the FactoMineR package and ggplot2 package in R software (Version 2.15.3). Principal coordinate analysis (PCoA) was then performed to obtain the principal coordinates and visualize the results from complex, multidimensional data. A distance matrix of weighted or unweighted UniFrac samples was transformed to a new set of orthogonal axes, by which the maximum variation factor was demonstrated by the first principal coordinate, and the second maximum one by the second principal coordinate, and so on. PCoA analysis graphs were generated with the WGCNA stat package and ggplot2 package in R software (Version 2.15.3). Unweighted pair-group method with arithmetic means (UPGMA) clustering was performed as a type of hierarchical clustering method to interpret the distance matrix using the average linkage and was conducted by QIIME software (version 1.7.0) [39].

### 2.10. Statistical Analysis

Statistical analysis of the non-sequencing data was performed using SPSS 21.0 statistical software (IBM, Chicago, IL, USA). ANOVA and Tukey’s test were applied to compare the statistical differences in the periodontal bacteria/total bacteria (%) and cytokine expression levels among the groups. Data are represented as either the mean, mean ± SD or median ± 95% confidence interval (CI), as noted in every figure legend. Values of *p* ≤ 0.05 were considered significant.

## 3. Results

### 3.1. Oral Periodontal Pathogens Are Present in Female Upper Reproductive Organs, Plus Heart, and Aorta, Indicating Potential Hematogenous Transmission; Nisin Mitigates Changes in the Oral Cavity

A PCR-based approach was used to evaluate the presence of periodontal pathogens in the oral cavity, heart, aorta and female upper reproductive organs. The oral swabs revealed that *P. gingivalis*, *T. forsythia*, and *F. nucleatum* were present at significantly higher levels in the infection group compared to the control group (*p* < 0.05) (Figure 2). Nisin treatment significantly mitigated the high levels of these pathogens (*P. gingivalis*, *T. forsythia*, and *F. nucleatum*) in the oral cavity compared to the infection group.

We also observed that *P. gingivalis* and *T. forsythia* were detected in the uterus, ovary, vagina, heart and aorta in the control and polymicrobial periodontal infection groups (Figure 2). Compared with the control group, the level of *P. gingivalis* in the uterus was significantly higher in the polymicrobial periodontal infection group. *T. denticola* and *F. nucleatum* were not detected in the uterus, ovary and vagina after polymicrobial periodontal infection. A lower amount of *F. nucleatum* was detected in the heart and aorta compared to the oral swabs at the eighth week.

The detection frequencies of the four bacteria at the eighth week in the oral swabs, uterus, ovary, vagina, heart and aorta are shown in Table 1. *P. gingivalis* and *T. forsythia* were detected with high frequency (≥50%) in all the organ sites and for all the groups. *F. nucleatum* was detected with the highest frequency (100%) in the oral cavity but with low frequency (≤50%) in the heart and aorta. *T. denticola* was only detected in the oral cavity and with lower frequency (≥50%) compared to other periopathogens (100% frequency). No significant difference was found for any frequency of the groups.

### 3.2. Polymicrobial Periodontal Infection Triggers an Increase in TNF-α and IL-6 Expression in the Oral Cavity and Uterus; Nisin Mitigates These Changes

The polymicrobial periodontal infection group had a significantly higher TNF-α expression in the uterus and a higher trend in the oral cavity compared to the control group (Figure 3). The periodontal infection also mediated a significant increase in IL-6 expression in the oral cavity and uterus compared to the control. The nisin treatment groups had lower cytokine expression levels (TNF-α and IL-6) in the oral cavity and uterus compared to the infection groups. Although there was a trend toward increased expression levels for IL-1b in the oral cavity of the infection group, this did not reach statistical significance. There were no significant differences in IL-1b expression between the control and infection groups for any organs.

### 3.3. Polymicrobial Periodontal Infection Alters the Microbiome Alpha and Beta Diversity of the Female Reproductive System and Nisin Mediates Changes

The polymicrobial periodontal infection groups had significantly lower alpha diversity indices, namely the Shannon index and Simpson index, in the uterus compared to the controls (Figure 4). In the context of infection, the nisin treatment group had higher Shannon and Simpson indices in the uterus compared with the control, although only the Shannon index reached statistical significance. Nisin treatment also showed a higher Simpson index and Shannon index in the vagina compared with the infection group. Although nisin treatment led to a significantly higher Chao1 index in the ovary and heart compared with the infection group, there were no significant differences between the control and infection groups. Nisin treatment alone or in the context of infection led to higher observed species in the uterus, ovary, vagina, and heart compared with the control group.

We next evaluated the beta diversity using principal component analysis (PCoA) of the OUT-level data, and the results can be seen in Figure 5. The analysis showed distinct groups between the control and infection groups, but these groups overlapped for all the organ sites. The nisin groups (nisin control and nisin plus infection) separated from the control and infection groups in the uterus and ovary and showed greater overlap in the vagina and heart.

### 3.4. Periodontal Infection Shifts the Relative Abundance and Microbial Composition in the Uterus, Ovary and Vagina; Nisin Mitigates Some Changes and Establishes a New State

We next evaluated the microbial relative abundance in the specific microbial composition for the different groups at the different organ sites. The results were divided into phyla- (Figure 6a and Figure 7) and genus-level (Figure 6b and Figure 8) data. At the phylum level, common bacterial phyla, such as *Firmicutes*, *Bacteroidetes*, *Proteobacteria*, *Actinobacteria* and *Cyanobacteria*, were dominant in the uterus, ovary, vagina and heart tissue. At the phylum level, the polymicrobial periodontal infection compared to the control group generally had lower *Firmicutes* and higher *Bacteroidota* and *Proteobacteria* in the reproductive organs compared with the control group (Figure 6a). However, nisin treatment alone or in the presence of infection generally had higher *Proteobacteria* and lower *Firmicutes* and *Bacteroidota* in the reproductive organs and the heart compared to the control or infection groups. In the vaginal tissue, *Firmicutes* were significantly higher in the control group compared to the infection group. Several findings reached statistical significance (Figure 7).

At the genus level, the polymicrobial periodontal infection groups generally had higher *Muribaculaceae* and lower *Lachnospiraceae NK4A136* group in all the reproductive organs and the heart compared with the control groups (Figure 6b and Figure 8). However, the nisin groups or infection+nisin groups had higher *Pseudomonas* and lower *Muribaculaceae* and *Lachnospiraceae NK4A136* groups in the reproductive organs and heart tissue compared to the infection groups (Figure 6b and Figure 8). Moreover, compared with the control group, the polymicrobial periodontal infection group had higher levels of the *Turicibacter* group in the reproductive organs and heart (Figure 8). Several findings reached statistical significance (Figure 8).

In summary, the polymicrobial infection generally triggered a lower microbiome diversity in the uterus, which was abrogated by nisin treatment. The polymicrobial infection generally had lower *Firmicutes* and higher *Bacteroidota* levels in all the reproductive organs compared to the control group, with similar trends revealed in the heart. However, nisin treatment alone or in the presence of infection generally had higher *Proteobacteria* and lower *Firmicutes* and *Bacteroidota* in the reproductive organs and the heart compared to the control or infection groups. Nisin treatment also altered the microbiome community structure in the reproductive tract to a new state that did not mirror the controls. Periodontal disease triggered higher levels of inflammatory cytokines (IL-6, TNF-α) in the uterus, which was abrogated by nisin treatment.

## 4. Discussion

The female reproductive tract harbors distinct microbial communities in the vagina, uterus, fallopian tubes and ovary. The nature of the vaginal microbiota has been well studied [41,42]. In contrast, the upper reproductive tract remains largely unexplored. A healthy reproductive tract microbiota mediates critical functions in the host, such as development of the immune system, protection against opportunistic infections, facilitation of digestion, and production of bioactive metabolites [6,43]. Alteration of the upper reproductive tract microbiota is likely to play an important role in pelvic inflammatory diseases, which are strongly associated with female reproduction [43]. In this study, a polymicrobial infection mouse model of periodontal disease was used to evaluate the effect of a periodontal infection on the female reproductive tract microbiota. Polymicrobial infection models of periodontal disease, unlike mono-microbial infection models, ligature models and injection models, are less acute and more closely recapitulate the chronic and systemic pathogenic nature of periodontal disease. Therefore, polymicrobial models are useful for studying the systemic impacts of periodontal disease at distant organ sites, like the reproductive tract.

We verified the colonization of periodontal pathogens in the oral cavity, uterus, ovary, vagina, heart and aorta tissue. The PCR results revealed that all four periodontal pathogens (*P. gingivalis*, *T. denticola*, *T. forsythia*, *F. nucleatum*) colonized the oral cavity of the infected mice, although *T. denticola* was less effective compared with the other three pathogens. However, of the four bacterial pathogens, only *P. gingivalis* and *T. forsythia* could be detected in the female reproductive organs (uterus, ovary and vagina) and three of the pathogens (*P. gingivalis*, *T. forsythia*, *F. nucleatum)* could be detected in the vascular sites (heart and aorta). This result suggests that *P. gingivalis* and *T. forsythia* may spread from the oral cavity to distal reproductive organs via hematogenous transmission. Furthermore, although *F. nucleatum* was detected at vascular sites (heart and aorta), *T. denticola* and *F. nucleatum* may not spread readily to the upper reproductive tract through the circulation. In addition, it is noteworthy that *P. gingivalis* and *T. forsythia* were detected in the uterus, ovary, vagina, heart and aorta in both the control and infection groups. There are a few possible explanations for this finding. The first explanation is that these microbes are native to rodents, as has been previously reported [20,25]. Furthermore, the BALB/cByJ mice were not germ-free mice and these periodontal pathogens might have normally colonized the reproductive organs before infection, albeit in low numbers. A second possibility is cross-contamination between groups; although the mice groups were kept in different cages, we still could not completely rule out environmental contamination. For this reason, all the mice were treated with systemic and local antibiotics/antimicrobials to eliminate bacterial load and carriage of contaminating species prior to the start of the study. Another reason might be the detection of a false positive due to using a very sensitive quantitative real-time PCR (qRT-PCR) method.

The current study also showed a higher host immune cytokine (TNF-α and IL-6) response in the oral cavity and uterus as a result of the polymicrobial infection compared to the CTL, indicating a local and systemic inflammatory response to the microbes and microbial products. Previous studies have reported that *P. gingivalis* and *T. forsythia* trigger an increase in the cytokine levels in different organs and serum. Lin et al. [44] used a subcutaneous chamber model and found that female mice challenged with heat-killed *P. gingivalis* revealed *P. gingivalis* DNA and an elevation in the TNF-α levels in the liver, uterus, spleen, and serum. *T. forsythia* was also reported to induce pro-inflammatory cytokines, such as IL-1β and IL-6 by CD4+ T helper cells, and TNF-α systemically [45,46].

The current data show that the polymicrobial infection groups had a significantly lower microbial diversity, especially in the Shannon diversity index. Similar results were found in several other animal studies. Wu et al. reported that increased *P. gingivalis* colonization was linked to reduced bacterial diversity in mice [47]. Our previous study found that infection with periodontal pathogens reduced the Shannon index in the oral cavity [18]. Nisin treatment increased the alpha diversity in the oral cavity and brain tissues [18,20].

At the genus level, the polymicrobial infection reduced the levels of the *Lachnospiraceae* NK4A136 group in the gingiva, uterus, ovary, vagina and heart microbiome. The *Lachnospiraceae* NK4A136 genus is comprised of a group of butyrate-producing bacteria, which have been reported in gut microbiome studies. In particular, the *Lachnospiraceae* NK4A136 group was found to maintain the gut barrier integrity in mice and inhibit inflammation [48]. Manuel et al. [49] reported that a high-fat diet decreased colonization of the gut by the *Lachnospiraceae* NK4A136 group, leading to a dysregulated immune response and subsequent spontaneous preterm birth. Ma et al. [50] reported that the *Lachnospiraceae* NK4A136 group was decreased in the gut microbiota of obese subjects. Moreover, the polymicrobial infection increased *Turicibacter* in the uterus, ovary, vagina and heart tissue. In recent studies, increased *Turicibacter* genera in the intestinal flora has been associated with abnormal metabolism in systemic diseases, such as depression, Parkinson’s disease, type 2 diabetes and obesity [51,52,53,54]. Thus, a polymicrobial periodontal infection appears to reduce systemic levels of health-promoting microbes, like the *Lachnospiraceae* NK4A136 genus, while increasing the levels of disease-associated microbes, like *Turicibacter.*

Nisin treatment exerted multiple effects in the context of the polymicrobial infection. Nisin treatment significantly decreased the levels of the periodontal pathogens, namely *P. gingivalis*, *T. forsythia*, and *F. nucleatum*, in the oral cavity. Nisin treatment also altered the microbiome composition, diversity, and community structure in the reproductive tract to a new state that did not mirror the controls. Periodontal disease triggered an increase in inflammatory cytokines (IL-6, TNF-α) in the uterus, which was abrogated by nisin treatment. Thus, nisin exerts multiple beneficial local and systemic effects in the context of a periodontal infection, although its impact on the reproductive microbiome warrants further exploration.

In summary, the data indicate that a polymicrobial periodontal infection may affect the female reproductive physiology by disturbing the female reproductive organs’ microbiome and increasing cytokine expression, and nisin mediates the beneficial effects. However, some limitations of the present study are that the findings are only applicable to mouse models of periodontal disease and the model was not conducted in a germ-free environment but only in a specific-pathogen-free environment.

## Figures and Tables

**Figure 1 microorganisms-12-01647-f001:**
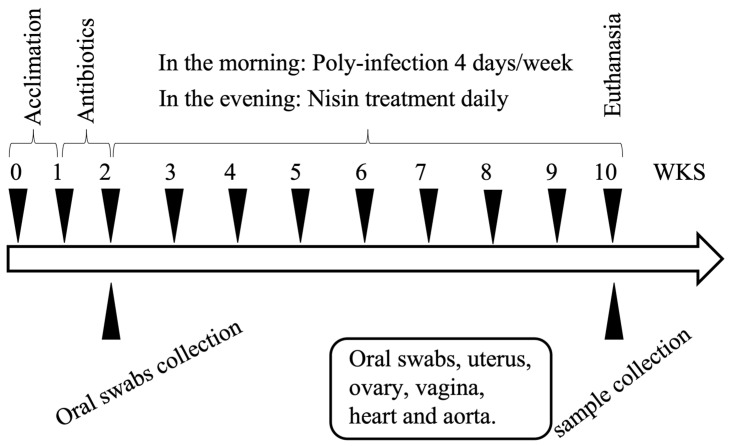
A diagram of the polymicrobial infection protocol. Following 1 week of acclimation and 1 week of antibiotic administration, the mice were infected with the polymicrobial infection from weeks (WKS) 2 to 10. Oral swab samples were collected at baseline after the antibiotic administration and at the end of the experimental period at 10 weeks. Tissues (uterus, ovary, vagina, heart, aorta) were also collected at 10 weeks.

**Figure 2 microorganisms-12-01647-f002:**
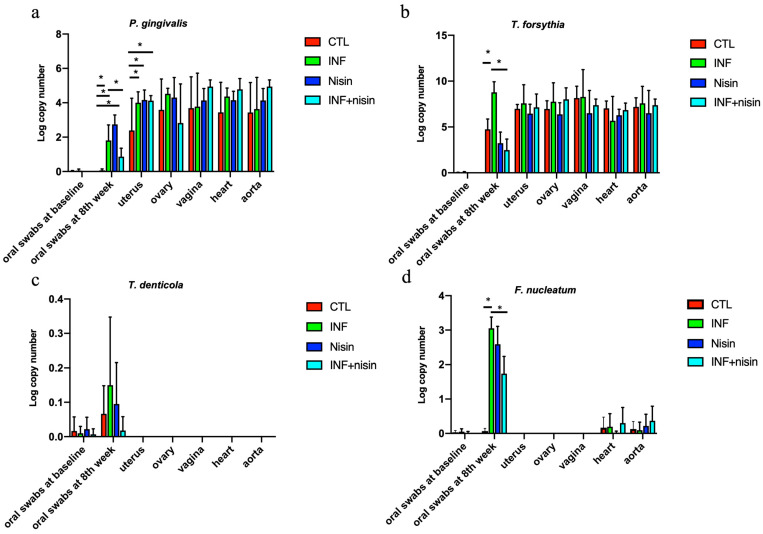
Oral periodontal pathogens are present in the female upper reproductive organs, plus the heart and aorta, and nisin mitigates changes in the oral cavity. The graphs show the log copy number of the periodontal pathogens *P. gingivalis* (**a**), *T. forsythia* (**b**), *T. denticola* (**c**), and *F. nucleatum* (**d**) in the oral swabs (at baseline and infection after 8 weeks) and uterus, ovary, vagina, heart and aorta tissue. Data are represented as the mean ± SD. * indicates significant differences between marked groups (*p* ≤ 0.05).

**Figure 3 microorganisms-12-01647-f003:**
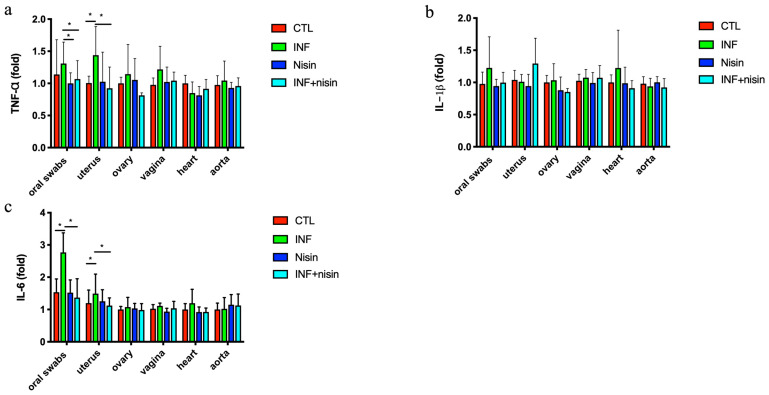
Polymicrobial periodontal infection triggers an increase in TNF-α and IL-6 mRNA expression in the oral cavity and uterus; nisin mitigates these changes. The graphs show the mRNA cytokine expression levels, including TNF-α (**a**), IL-β (**b**), and IL-6 (**c**), in the oral swabs and uterus, ovary, vagina, heart and aorta. Data are represented as the mean ± SD. * indicates significant differences between the two marked groups (*p* ≤ 0.05).

**Figure 4 microorganisms-12-01647-f004:**
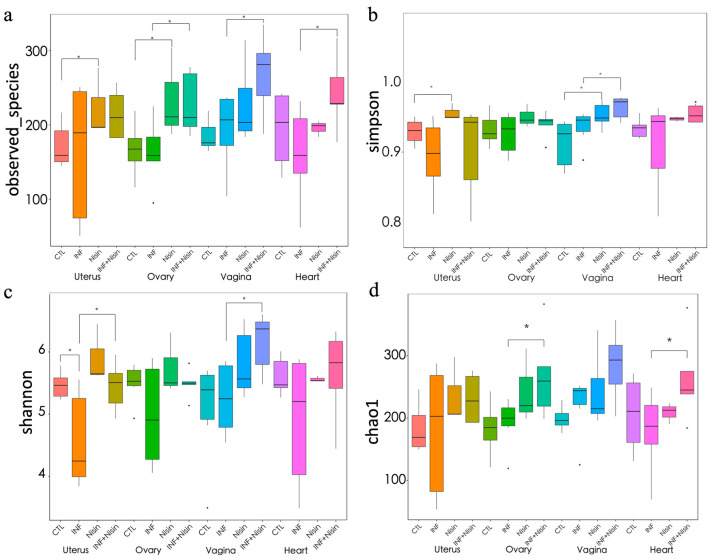
Polymicrobial periodontal infection alters the microbiome alpha diversity of the female reproductive system and nisin mediates the changes. The graphs show the alpha diversity of the observed species (**a**) and the Simpson (**b**), Shannon (**c**), and Chao1 (**d**) indices for the uterus, ovary, vagina, and heart tissue. Data are represented as the median ± 95% CI. * indicates significant differences between the two marked groups (*p* ≤ 0.05).

**Figure 5 microorganisms-12-01647-f005:**
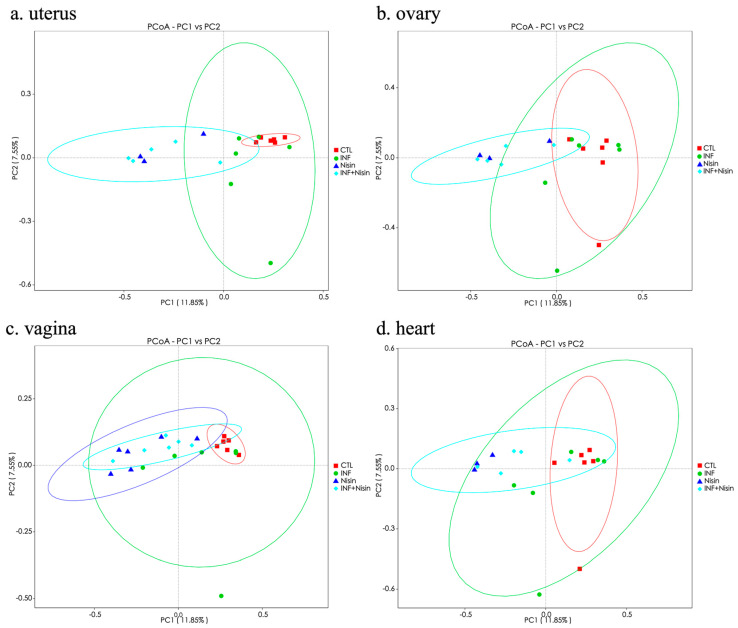
Polymicrobial periodontal infection tends to alter the microbiome beta diversity of the female reproductive system and nisin mediates the changes. Principal coordinate analysis (PCoA) representing the beta diversity for the uterus (**a**), ovary (**b**), vagina (**c**), and heart (**d**).

**Figure 6 microorganisms-12-01647-f006:**
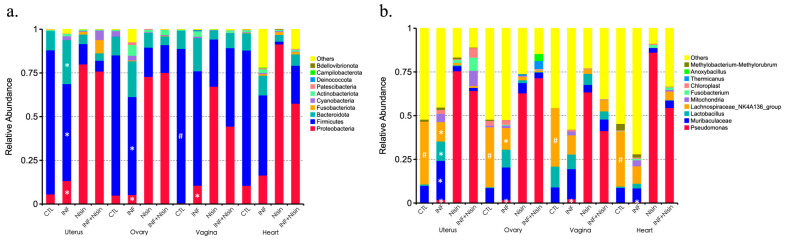
Periodontal infection shifts the relative abundance and microbial composition in the uterus, ovary and vagina; nisin mitigates some changes and establishes a new state. Bar graphs show the relative abundance of each bacteria taxa at the phylum level (**a**) and genus level (**b**) in the oral cavity, uterus, ovary, vagina and heart. Data are represented as the mean. # indicates a significant difference between the control and infection groups (*p* ≤ 0.05). * indicates a significant difference between the infection and infection+nisin groups (*p* ≤ 0.05).

**Figure 7 microorganisms-12-01647-f007:**
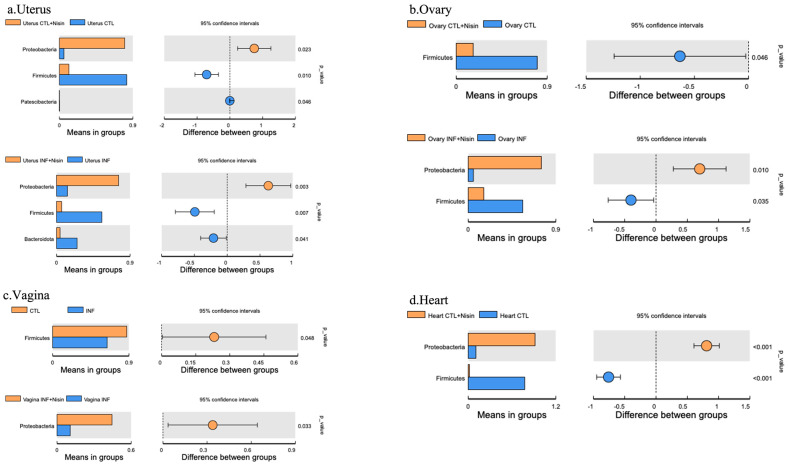
Periodontal infection significantly shifts the relative abundance and microbial phylum composition in the uterus, ovary and vagina; nisin mitigates some changes and establishes a new state. Bar graphs show the significant microbial differences between the groups at the phylum level in the uterus (**a**), ovary (**b**), vagina (**c**), and heart (**d**). Data are represented as the means in the bar graphs on the left, and as the 95% CIs in the scales on the right; *p* values are shown in each figure for each species.

**Figure 8 microorganisms-12-01647-f008:**
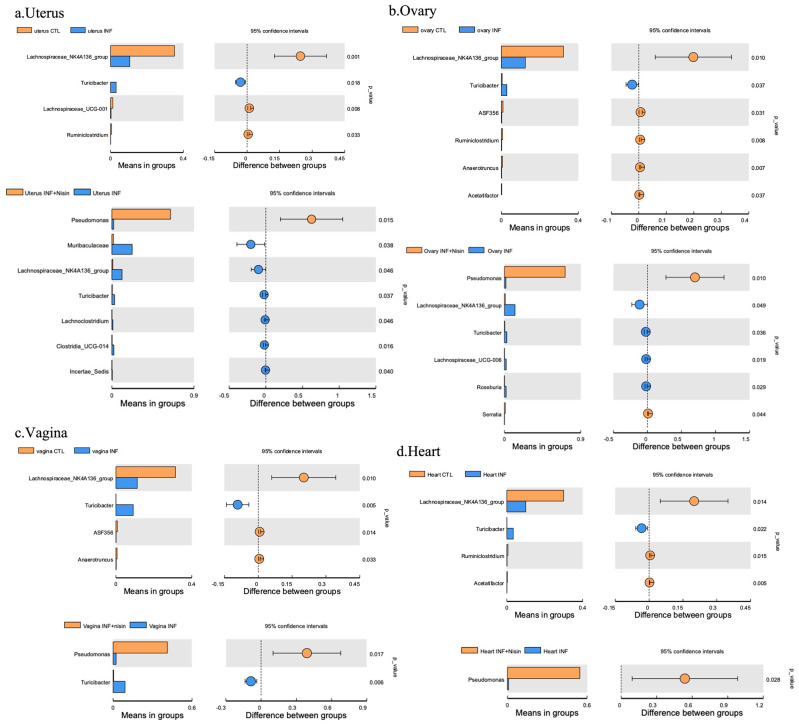
Periodontal infection significantly shifts the relative abundance and microbial genus composition in the uterus, ovary and vagina; nisin mitigates some changes and establishes a new state. Bar graphs show the significant microbial differences between the groups at the phylum level in the uterus (**a**), ovary (**b**), and vagina (**c**) and heart (**d**). Data are represented as the means in the bar graphs on the left, and as the 95% CIs in the scales on the right; *p* values are shown in each figure for each species.

**Table 1 microorganisms-12-01647-t001:** Detection frequency of the periodontal pathogens in the oral swabs, uterus, ovary, vagina, heart, and aorta. Values are expressed as the percentage (%) of total mice and with the total frequency in parenthesis (N = 6 per group).

	Oral Swab				Uterus				Ovary				Vagina				Heart				Aorta			
	CTL	INF	Nisin	INF+Nisin	CTL	INF	Nisin	INF+Nisin	CTL	INF	Nisin	INF+Nisin	CTL	INF	Nisin	INF+Nisin	CTL	INF	Nisin	INF+Nisin	CTL	INF	Nisin	INF+Nisin
** *P. gingivalis* **	100.0 (6/6)	100.0 (6/6)	100.0 (6/6)	100.0 (6/6)	66.7 (4/6)	100.0 (6/6)	100.0 (6/6)	100.0 (6/6)	50.0 (3/6)	100.0 (6/6)	100.0 (6/6)	66.7 (4/6)	100.0 (6/6)	100.0 (6/6)	100.0 (6/6)	100.0 (6/6)	83.3 (5/6)	100.0 (6/6)	100.0 (6/6)	100.0 (6/6)	83.3 (5/6)	83.3 (5/6)	100.0 (6/6)	100.0 (6/6)
** *T. denticola* **	50.0 (3/6)	50.0 (3/6)	66.7 (4/6)	16.7 (1/6)	0.0 (0/6)	0.0 (0/6)	0.0 (0/6)	0.0 (0/6)	0.0 (0/6)	0.0 (0/6)	0.0 (0/6)	0.0 (0/6)	0.0 (0/6)	0.0 (0/6)	0.0 (0/6)	0.0 (0/6)	0.0 (0/6)	0.0 (0/6)	0.0 (0/6)	0.0 (0/6)	0.0 (0/6)	0.0 (0/6)	0.0 (0/6)	0.0 (0/6)
** *T. forsythia* **	100.0 (6/6)	100.0 (6/6)	100.0 (6/6)	100.0 (6/6)	100.0 (6/6)	100.0 (6/6)	100.0 (6/6)	100.0 (6/6)	100.0 (6/6)	100.0 (6/6)	100.0 (6/6)	100.0 (6/6)	100.0 (6/6)	100.0 (6/6)	100.0 (6/6)	100.0 (6/6)	100.0 (6/6)	83.3 (5/6)	100.0 (6/6)	100.0 (6/6)	100.0 (6/6)	100.0 (6/6)	100.0 (6/6)	100.0 (6/6)
** *F. nucleatum* **	100.0 (6/6)	100.0 (6/6)	100.0 (6/6)	100.0 (6/6)	0.0 (0/6)	0.0 (0/6)	0.0 (0/6)	0.0 (0/6)	0.0 (0/6)	0.0 (0/6)	0.0 (0/6)	0.0 (0/6)	0.0 (0/6)	0.0 (0/6)	0.0 (0/6)	0.0 (0/6)	33.3 (2/6)	33.3 (2/6)	0.0 (0/6)	16.7 (1/6)	33.3 (2/6)	16.7 (1/6)	33.3 (2/6)	50.0 (3/6)

## Data Availability

The data that support the findings of this study are openly available in Figshare at http://www.doi.org/10.6084/m9.figshare.25124210.

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
