# Peer review of "Nisin, a Probiotic Bacteriocin, Modulates the Inflammatory and Microbiome Changes in Female Reproductive Organs Mediated by Polymicrobial Periodontal Infection"

_microorganisms, 2024, doi:10.3390/microorganisms12081647_

Round 1

Reviewer 1 Report (Previous Reviewer 1)

Comments and Suggestions for Authors

The authors have adequately responded to my comments, corrected errors in their figures and tables, and substantially improved the description of their methods.

Author Response

Thank you so much.

Best

Changchang Ye

Reviewer 2 Report (Previous Reviewer 3)

Comments and Suggestions for Authors

Authors revised the manuscript, but this article has not fully answered some of the questions due to inadequate statistical analysis and insufficient description.

First, as mentioned in the previous review, authors suggest “Although only P. gingivalis was significantly elevated in the oral cavity and uterus, and T. forsythia and F. nucleatum were significantly elevated in the oral cavity” (L32), “The polymicrobial infection compared to the control generally decreased the Firmicutes and increased the Bacteroidota in all reproductive organs with similar trends revealed in the heart.” (L38), “nisin treatment alone or in the presence of infection compared to the control or infection groups generally increased the Proteobacteria and reduced the Firmicutes and Bacteroidota in reproductive organs and the heart.” (L40), “Periodontal disease triggered an increase in inflammatory cytokines (IL-6, TNF-) in the uterus” (L43), “The polymicrobial periodontal infection group had a significantly higher TNF- expression in the uterus and an increasing trend in the oral cavity compared to control (Figure 3).” (L297), “Nisin treatment mitigated this increased cytokine expression in TNF- and IL-6 in the oral cavity and uterus in the infection group.” (L300), “The polymicrobial periodontal infection significantly reduced the alpha diversity indices, namely the Shannon index and Simpson index, in the uterus compared to the controls (Figure 4).” (L314), “although only the change in the Shannon index reached statistical significance.” (L317), “We next evaluated the microbial relative abundance and changes in specific microbial composition for the different treatment groups at the different organ sites.” (L345), “At the genus level, the polymicrobial periodontal infection generally increased the Muribaculaceae and decreased the lachnospiraceae NK4A136 group in all reproductive organs and the heart compared with the control group (Figure 6b, 8).” (L372), “nisin treatment alone or in the context of infection increased the Pseudomonas and decreased the Muribaculaceae, and Lachnospiraceae NK4A136 groups in the reproductive organs and heart tissue compared to infection (Figure 6b, 8).” (L374), and “The current data show that the polymicrobial infection induces a significant reduction in microbial diversity” (L447), but they do not show the results of statistical analysis (i.e., the difference between “before” and “after”). Authors could not use the words such as “increased” and “reduction”, because they do not examine them before intervention. Authors should describe the results sections based on the results.

Second, authors suggest “Nisin treatment significantly decreased the levels of P. gingivalis, T. forsythia, and F. nucleatum in the oral cavity.” (L34), but they do not show statistical results of difference between before and after intervention for oral swabs in figure 2. If authors need to explain so, they should show statistical results of the differences between before and after intervention.

Author Response

First, as mentioned in the previous review, authors suggest “Although only P. gingivalis was significantly elevated in the oral cavity and uterus, and T. forsythia and F. nucleatum were significantly elevated in the oral cavity” (L32), “The polymicrobial infection compared to the control generally decreased the Firmicutes and increased the Bacteroidota in all reproductive organs with similar trends revealed in the heart.” (L38), “nisin treatment alone or in the presence of infection compared to the control or infection groups generally increased the Proteobacteria and reduced the Firmicutes and Bacteroidota in reproductive organs and the heart.” (L40), “Periodontal disease triggered an increase in inflammatory cytokines (IL-6, TNF-a) in the uterus” (L43), “The polymicrobial periodontal infection group had a significantly higher TNF-a expression in the uterus and an increasing trend in the oral cavity compared to control (Figure 3).” (L297), “Nisin treatment mitigated this increased cytokine expression in TNF-a and IL-6 in the oral cavity and uterus in the infection group.” (L300), “The polymicrobial periodontal infection significantly reduced the alpha diversity indices, namely the Shannon index and Simpson index, in the uterus compared to the controls (Figure 4).” (L314), “although only the change in the Shannon index reached statistical significance.” (L317), “We next evaluated the microbial relative abundance and changes in specific microbial composition for the different treatment groups at the different organ sites.” (L345), “At the genus level, the polymicrobial periodontal infection generally increased the Muribaculaceae and decreased the lachnospiraceae NK4A136 group in all reproductive organs and the heart compared with the control group (Figure 6b, 8).” (L372), “nisin treatment alone or in the context of infection increased the Pseudomonas and decreased the Muribaculaceae, and Lachnospiraceae NK4A136 groups in the reproductive organs and heart tissue compared to infection (Figure 6b, 8).” (L374), and “The current data show that the polymicrobial infection induces a significant reduction in microbial diversity” (L447), but they do not show the results of statistical analysis (i.e., the difference between “before” and “after”). Authors could not use the words such as “increased” and “reduction”, because they do not examine them before intervention. Authors should describe the results sections based on the results.

-Thank you for your comment. We changed these sentences.

Second, authors suggest “Nisin treatment significantly decreased the levels of P. gingivalis, T. forsythia, and F. nucleatum in the oral cavity.” (L34), but they do not show statistical results of difference between before and after intervention for oral swabs in figure 2. If authors need to explain so, they should show statistical results of the differences between before and after intervention.

- Thank you for your comment. According to our protocol/methods, we infected and treated mice on the same day (see Figure 1); our experimental protocol design did not utilize a before and after treatment design protocol for the same group. We have modified the mentioned sentence.

Round 2

Reviewer 2 Report (Previous Reviewer 3)

Comments and Suggestions for Authors

I have no further comment.

This manuscript is a resubmission of an earlier submission. The following is a list of the peer review reports and author responses from that submission.

Round 1

Reviewer 1 Report

Comments and Suggestions for Authors

The work entitled: Nisin, a Probiotic Bacteriocin, Modulates Inflammatory and Microbiome Changes in Female Reproductive Organs Mediated by Polymicrobial Periodontal Infection; determines the effect of the Nisin polypeptide on the microbiota of the uterus, ovary, vagina, heart, and aorta.

They perform a controlled infection with four bacterial species, P. gingivalis, T. denticola, T. forsythia, and F. nucleatum, using an oral gavage and a treatment with the bacteriocin Nisin.

They demonstrate that the addition of bacteriocin modifies the prevalence of some of the coinfected strains in some organs and that the microbiota is changing globally, with a strong increase in the presence of Proteobacteria, particularly Pseudomonas.

The work is well written, its experiments are clear, and the statistical analysis is adequate, although its data show marginal effects for the control of coinfected bacteria.

For its acceptance, the work requires addressing some points listed below:

1) Explain what happens with Treponema denticola in the presence of Nisin; it is no longer mentioned in the summary

2) In Fig 2, it seems that Nisin only influences the oral cavity; is that correct?

3) Figure 8, unlike the other figures, has the figure caption at the top of the image; the presentation needs to be standardized

4) Why in the table is F. nucleatum more likely to be found when INF + Nisin is administered than when INF alone? (50 vs 16.7). Something similar in T. forsythia in the heart (100 vs 83.3) and P. gingivalis in the aorta (100 vs 83.3)

5) I did not find the figure caption for Table 1

6) Using the same color code as in Figure 2 in Figure 3 would be advisable.

7) In some TNF- of the text, the  is missing

8) In the text, they repeat Figure 4 as Figure 3, which is different from the attached material

9) Figure 5 only has panels a, b, c, and d, and panel e is mentioned in the text. The figure caption needs to be fixed.

Author Response

Dear Reviewer:

Thank you so much for your kind suggestions and comment. Your suggestions help us improved our manuscript. We corrected our manuscript followed your advice.

1) Explain what happens with Treponema denticola in the presence of Nisin; it is no longer mentioned in the summary

Thank you for your comment. We had 6 mice in one group. In Nisin group, Treponema denticola were detected in 4 mice compared with INF group(3 mice). First, the current study used the more sensitive quantitative real-time PCR (qRT-PCR) method of detection. Thus, we may have detected a false positive. Second, we cannot ignore the low level contamination between the mice. Because there were 6 mice live in one cage. Although, all mice were treated with antibiotics to eliminate bacterial load and carriage of contaminating species prior to the start of the study.

2) In Fig 2, it seems that Nisin only influences the oral cavity; is that correct?

Thank you for your comment. Yes, nisin reduced the periodontal bacteria in oral cavity. Nisin also regulated cytokine expression in uterus.

3) Figure 8, unlike the other figures, has the figure caption at the top of the image; the presentation needs to be standardized

Thank you for your advice. Since the version is offered by editors. They will help us to change the order of figure and caption.

4) Why in the table is F. nucleatum more likely to be found when INF + Nisin is administered than when INF alone? (50 vs 16.7). Something similar in T. forsythia in the heart (100 vs 83.3) and P. gingivalis in the aorta (100 vs 83.3)

Thank you for your comment. The nisin changed the composition of the microbiome which is more suitable for F. nucleatum.

5) I did not find the figure caption for Table 1

Thank you for your advice, we added the caption for table 1:Table 1 Detection frequency of periodontal pathogens in the oral swabs, uterus, ovary, vagina, heart, and aorta; the total number mice in each group was 6. The percentage of detection equals bacteria positive mice number /mice in each group.

6) Using the same color code as in Figure 2 in Figure 3 would be advisable.

Thank you for your suggestion, yes we used the same color code.

7) In some TNF-a of the text, the a is missing

Thank you for your advice. The new version has some problems in showing alpha. Editor will correct it.

8) In the text, they repeat Figure 4 as Figure 3, which is different from the attached material

Thank you for your advice. The new version has some problems in showing data. Editor had corrected.

9) Figure 5 only has panels a, b, c, and d, and panel e is mentioned in the text. The figure caption needs to be fixed.

Thank you for your comment. We changed the caption.

Reviewer 2 Report

Comments and Suggestions for Authors

The present study examined the effects of a probiotic bacteriocin, nisin, in modulating the reproductive microbiome and inflammation triggered by periodontitis.   A polymicrobial mouse model of periodontal disease was used to evaluate the effects of this disease on the female reproductive system with a focus on the microbiome, local inflammation, and nisin’s therapeutic potential. The authors conclude that nisin treatment alters the reproductive tract microbial profile, microbiome, and inflammatory status induced by the polymicrobial periodontal infection.

Taken together the findings in this paper might contribute to explain unexpected infertility and the pre-term birth issue associated with periodontitis during or before pregnancy. In addition, nisin seems to be a promising therapeutic agent to use for to treat this problem.

General aspects:

The manuscript is well-written and describes an important topic for research.

Taken together, the study model is relevant and the selected analyses of oral and systemic distribution of bacterial DNA, as well as the cytokines target for quantifications. However, pregnant women are young individuals in regard to the periodontitis populations. In young individuals the presence of the facultatively periodontal pathogen Aggregatibacter actionomycetemcomitans is high. In addition, cultivation techniques for detection of viable systemic bacteria would have further strengthen the impact of the study.

Specific points:

Line 112          Suggest the authors should avoid the word “probiotic”, which can mislead the reader in the present study.

Line-121-130  Only strict anaerobic bacteria were selected for the study. Do they survive and grow in mouse model?

Line 155          How was the polymicrobial periodontal infection administrated? Oral, in the drinking water.

Line 161          Was administration of Nisin similar as for the antibiotic, in the drinking water?

Line 191          The normalization of the relative cytokine expression could be better explained.

Line 200-224 What was the threshold for detection of the bacterial DNA (genome(bacteria)/g (sample)? Was it the same for all 4 periodontal pathogens?

Line 200-224 Is there a suggested route for the translocation of bacterial DNA to the systemic organs? This is amazing data, and therefore needs of a solid explanation.

Line 250-257 High levels of the bacteria? The detection threshold and high levels needs to be explained.

Line 356-357 I suggest a result summary et the end of the result section. Data presented is extensive and needs to be concluded.

               How could nisin be administrated in a clinical trial. By probiotic therapy or by giving the protein in a solution.

              Could the result of nisin be compared with that of a traditional antibiotic treatment.

              Is translocation of bacterial DNA a result of bacterial translocation or only DNA or vesicles? Can the authors provide with a hypothesis?

Author Response

Dear Reviewer:

Thank you so much for your kind suggestions and comment. Your suggestions help us improved our manuscript. We corrected our manuscript followed your advice.

Taken together, the study model is relevant and the selected analyses of oral and systemic distribution of bacterial DNA, as well as the cytokines target for quantifications. However, pregnant women are young individuals in regard to the periodontitis populations. In young individuals the presence of the facultatively periodontal pathogen Aggregatibacter actionomycetemcomitans is high.

Thank you for your comment. We choose four bacteria from Socransky’s red complex and orange complex which are more related with chronic periodontitis.

In addition, cultivation techniques for detection of viable systemic bacteria would have further strengthen the impact of the study. 

Thank you for your suggestion. Cultivation beyond the scope of this study.

Specific points:

Line 112          Suggest the authors should avoid the word “probiotic”, which can mislead the reader in the present study.

             The nisin is a production from probiotic L. lactis. We changed probiotic into bacteriocin.

Line-121-130  Only strict anaerobic bacteria were selected for the study. Do they survive and grow in mouse model?

            Yes, we think they survived. In our previous studies, periodontal pathogens induced host immune response in gingiva, liver and brain. These bacteria DNA were also detected in liver and brain. 

Line 155          How was the polymicrobial periodontal infection administrated? Oral, in the drinking water.

Thank you for your comment. The mice oral cavity was rinsed with the polymicrobial inoculum.

Line 161          Was administration of Nisin similar as for the antibiotic, in the drinking water? 

Thank you for your comment. The oral cavity of treatment group was rinsed with by the nisin solution which was mixed with an equal volume of sterile 4% CMC.

Line 191          The normalization of the relative cytokine expression could be better explained.

Thank you for your suggestion. We added glyceraldehyde 3-phosphate dehydrogenase (GAPDH; Mm99999915_g1) was used as a housekeeping gene to normalize the amount of mRNA present in each reaction.

Line 200-224 What was the threshold for detection of the bacterial DNA (genome(bacteria)/g (sample)? Was it the same for all 4 periodontal pathogens?

Thank you for your comment. Standard RT-PCR was used to confirm the presence of the periodontal pathogens in the oral swabs and tissue. Tenfold serial dilutions of DNA of known concentration were used to construct standard curves for quantification of periodontal pathogens. The standard positive control from 1010/100ng,109/100ng,108/100ng,107/100ng,106 /100ng ,105 /100ng ,104 /100ng ,103 /100ng ,102 /100ng ,101 /100ng to negative control 0. If the concentration is too high out of the range, we diluted the sample and test it again. 4 periodontal pathogens have different original standard concentrations.

Line 200-224 Is there a suggested route for the translocation of bacterial DNA to the systemic organs? This is amazing data, and therefore needs of a solid explanation.

         Thank you for your comment. In patients with periodontitis, the periodontal bacteria might enter into the blood circulation by transient bacteremia.

Line 250-257 High levels of the bacteria? The detection threshold and high levels needs to be explained.

Thank you for your comment. PCR positive standard curve the positive control decided the threshold. The lowest is 0. The highest is 1010.

Line 356-357 I suggest a result summary et the end of the result section. Data presented is extensive and needs to be concluded.

Thank you for your comment. We add: The polymicrobial infection generally decreased microbiome diversity in the uterus, which was abrogated by nisin treatment. The polymicrobial infection compared to the control generally decreased the Firmicutes and increased the Bacteroidota in all reproductive organs with similar trends revealed in the heart. However, nisin treatment alone or in the presence of infection compared to the control or infection groups generally increased the Proteobacteria and reduced the Firmicutes and Bacteroidota in reproductive organs and the heart. Nisin treatment also altered the microbiome community structure in the reproductive tract to a new state that did not mirror the controls. Periodontal disease triggered an increase in inflammatory cytokines (IL-6, TNF-a) in the uterus, which was abrogated by nisin treatment.

How could nisin be administrated in a clinical trial. By probiotic therapy or by giving the protein in a solution.

 Could the result of nisin be compared with that of a traditional antibiotic treatment.

Thank you for your comment. UCSF started the nisin clinical trail recently. Nisin was given by the protein in a solution.

Is translocation of bacterial DNA a result of bacterial translocation or only DNA or vesicles? Can the authors provide with a hypothesis?

Thank you for your comment. Further study needed to examine this.

Reviewer 3 Report

Comments and Suggestions for Authors

Authors used an animal experimental study to examine the effect of nisin on prevention of infection in reproductive organs. However, this article has not fully answered some of the questions due to inadequate statistical analysis and insufficient description.

First, authors suggest “The detection frequency of the four bacteia at the 8th week in the oral swabs, uterus, ovary, vagina, heart and aorta are shown in Table 1.” (L271), but they do not show the table 1 in the manuscript. Moreover, figure 3 may be the same as figure 4, and it is difficult to understand what they did for L278-L285. Authors should revise the manuscript, carefully.

Second, authors used “*”, boxes and bars in figure 2, but they do not explain what they were in figure legend. Moreover, authors suggest “The oral swabs revealed that P. gingivalis, T. forsythia, and F. nucleatum were present at significantly higher levels in the infection group compared to the control group (p < 0.001) (Figure 2).” (L251), but there is not “*” between them for F. nucleatum. Authors should rewrite the manuscript, carefully.

Third, authors suggest “Similarly, T. denticola showed a trend toward higher levels in the infection group” (L253), but as the sample size in each group is only 4, it is difficult to conclude the association as “trend”. Authors should rewrite the results based on results of the statistical analysis.

Fourth, authors suggest “Nisin treatment significantly mitigated the high levels of these pathogens (P. gingivalis, T. forsythia, and F. nucleatum) in the oral cavity.” (L255), but there were not * between the infection group and the infection + nisin group for P. gingivalis and F. nucleatum. Moreover, authors suggest “P. gingivalis colonization in the uterus was significantly increased after the polymicrobial periodontal infection.” (L265), but they do not show the results of statistical analysis (i.e., the difference between “before” and “after”). Furthermore, authors suggest “nisin trended to decrease the levels of P. gingivalis in the ovary and T. forsythia in the uterus and vagina” (L268), “nisin treatment increased both the Shannon and Simpson index in the uterus” (L294), “only the change in the Shannon index reached statistical significance” (L295), “nisin treatment compared to infection significantly elevated the Chao index in the ovary and heart.” (L299), “Nisin treatment alone or in the context of infection increased the observed species in the uterus, ovary, vagina, and heart.” (L300), “the polymicrobial periodontal infection increased the Turicibacter group in the reproductive organs and heart (Figure 8).” (L349), “The current study also showed an elevated host immune cytokine (TNF- and IL-6) response in the oral cavity and uterus” (L395), but they do not show the levels before the interventions, and therefore it is difficult to conclude the change of levels (e.g., “decrease”, “increased”, “change” and “elevated”). Authors should describe the results sections based on the results.

Fifth, authors used Shannon index, Simpson index and Chao index in figure 4 and OTU level data in figure 5, but they do not explain these indexes with reference in method section. It is difficult for readers to understand what authors did in the manuscript without the explanation. Authors should add explanations of these indicators in method section.

Sixth, authors showed the results in aorta in figure 2, but they do not show them in figure 3-8. It is difficult for reader to understand why author do not show the results of aorta without explanations. Authors should add explanations to justify why authors do not show the results of aorta in figure 3-8.

Seventh, authors suggest “The polymicrobial inoculum was administered topically in the morning for 4 consecutive days every week for a total of 8 weeks.” (L155), but they do not justify why the polymicrobial inoculum was not administered for every day. It is crucial to justify what authors did as intervention in experimental studies. Authors should add descriptions for the justification.

Eighth, authors do not show design of their study (e.g., the number of samples size in each group) in the abstract. Without information of the design, it is difficult for readers to understand what authors did in this study from the abstract. Authors should add descriptions of the design in abstract.

Finally, authors described some of sentences without citation or justification as follows; “the NCBI nonredundant database with DIAMOND to taxonomically annotate each metagenomic homolog (MEGAN).” (L221), “The nature of the vaginal microbiota has been well-studied.” (L359), and “Healthy reproductive tract microbiota mediate critical functions in the host, such as development of the immune system, protection against opportunistic infections, facilitation of digestion and production of bioactive metabolites.” (L360), but it is difficult for readers to judge it without references as evidence for each description. Authors should add references for these descriptions.

Minor comments

L179: “RNAlater solution” may be “RNA later solution”.

L179: “at 4 ºC” may be “at 4 ºC”.

L185: “(IL-1 )” may be “(IL-1)”.

L378: “the cardiovascular system (heart and aorta)” may be “vascular sites (heart and aorta)”.

L381: “vascular sites (heart, aorta)” may be “vascular sites (heart and aorta)”.

Comments on the Quality of English Language

Minor comments

L179: “RNAlater solution” may be “RNA later solution”.

L179: “at 4 ºC” may be “at 4 ºC”.

L185: “(IL-1 )” may be “(IL-1)”.

L378: “the cardiovascular system (heart and aorta)” may be “vascular sites (heart and aorta)”.

L381: “vascular sites (heart, aorta)” may be “vascular sites (heart and aorta)”.

Author Response

Dear Reviewer:

Thank you so much for your kind suggestions and comment. Your suggestions help us improved our manuscript. We corrected our manuscript followed your advice.

First, authors suggest “The detection frequency of the four bacteia at the 8th week in the oral swabs, uterus, ovary, vagina, heart and aorta are shown in Table 1.” (L271), but they do not show the table 1 in the manuscript. Moreover, figure 3 may be the same as figure 4, and it is difficult to understand what they did for L278-L285. Authors should revise the manuscript, carefully.

Thank you for your advice, there are some mistakes in editor’s version. The figure and table were corrected.

Second, authors used “*”, boxes and bars in figure 2, but they do not explain what they were in figure legend.

Thank you for your comment, we add * means significantly difference in figure 2 legend.

Moreover, authors suggest “The oral swabs revealed that P. gingivalis, T. forsythia, and F. nucleatum were present at significantly higher levels in the infection group compared to the control group (p < 0.001) (Figure 2).” (L251), but there is not “*” between them for F. nucleatum. Authors should rewrite the manuscript, carefully.

Thank you for your comment, we add * in F. nucleatum part figure2.

Third, authors suggest “Similarly, T. denticola showed a trend toward higher levels in the infection group” (L253), but as the sample size in each group is only 4, it is difficult to conclude the association as “trend”. Authors should rewrite the results based on results of the statistical analysis.

Thank you for your comment, We have 6 mice in each group. We added not significant after Trend.

Fourth, authors suggest “Nisin treatment significantly mitigated the high levels of these pathogens (P. gingivalis, T. forsythia, and F. nucleatum) in the oral cavity.” (L255), but there were not * between the infection group and the infection + nisin group for P. gingivalis and F. nucleatum.

Thank you for your comment, we add * in figure2.

Moreover, authors suggest “P. gingivalis colonization in the uterus was significantly increased after the polymicrobial periodontal infection.” (L265), but they do not show the results of statistical analysis (i.e., the difference between “before” and “after”).

Thank you for your comment, we do not have before and after data, We have INF group compared to control group

Furthermore, authors suggest “nisin trended to decrease the levels of P. gingivalis in the ovary and T. forsythia in the uterus and vagina” (L268), “nisin treatment increased both the Shannon and Simpson index in the uterus” (L294), “only the change in the Shannon index reached statistical significance” (L295), “nisin treatment compared to infection significantly elevated the Chao index in the ovary and heart.” (L299), “Nisin treatment alone or in the context of infection increased the observed species in the uterus, ovary, vagina, and heart.” (L300), “the polymicrobial periodontal infection increased the Turicibacter group in the reproductive organs and heart (Figure 8).” (L349), “The current study also showed an elevated host immune cytokine (TNF- and IL-6) response in the oral cavity and uterus” (L395), but they do not show the levels before the interventions, and therefore it is difficult to conclude the change of levels (e.g., “decrease”, “increased”, “change” and “elevated”). Authors should describe the results sections based on the results.

Thank you for your suggestion. We compared to CTL group.

Fifth, authors used Shannon index, Simpson index and Chao index in figure 4 and OTU level data in figure 5, but they do not explain these indexes with reference in method section. It is difficult for readers to understand what authors did in the manuscript without the explanation. Authors should add explanations of these indicators in method section.

Thank you for your comment, we add alpha and beta diversity analysis in material and methods part.

Sixth, authors showed the results in aorta in figure 2, but they do not show them in figure 3-8. It is difficult for reader to understand why author do not show the results of aorta without explanations. Authors should add explanations to justify why authors do not show the results of aorta in figure 3-8.

Thank you for your comment. Please refer to line198: The DNA from the uterus, ovary, vagina and heart met quality control measures for subsequent 16s RNA sequencing analysis. However, the DNA from aorta is limited for subsequent 16s RNA sequencing analysis.The aorta tissue was very limited. The quantity of aorta DNA was not enough after PCR and cytokine examinations.

Seventh, authors suggest “The polymicrobial inoculum was administered topically in the morning for 4 consecutive days every week for a total of 8 weeks.” (L155), but they do not justify why the polymicrobial inoculum was not administered for every day. It is crucial to justify what authors did as intervention in experimental studies. Authors should add descriptions for the justification. 

Thank you for your comment. We followed the infection method by other model including:

1.Chukkapalli, S. S. et al. Polymicrobial oral infection with four periodontal bacteria

orchestrates a distinct inflammatory response and atherosclerosis in apoe null

mice. PLoS ONE 10, e0143291 (2015).

2.Nahid, M. A., Rivera, M., Lucas, A., Chan, E. K. & Kesavalu, L. Polymicrobial infection with periodontal pathogens specifically enhances microRNA miR-146a in ApoE-/- mice during experimental periodontal disease. Infect. Immun. 79, 1597–1605 (2011).

3.Velsko, I. M. et al. Periodontal pathogens invade gingiva and aortic adventitia and elicit inflammasome activation in αvβ6 integrin-deficient mice. Infect. Immun. 83, 4582–93 (2015).

Eighth, authors do not show design of their study (e.g., the number of samples size in each group) in the abstract. Without information of the design, it is difficult for readers to understand what authors did in this study from the abstract. Authors should add descriptions of the design in abstract.

Thank you for your suggestion, we added the design into abstract.

Finally, authors described some of sentences without citation or justification as follows; “the NCBI nonredundant database with DIAMOND to taxonomically annotate each metagenomic homolog (MEGAN).” (L221), “The nature of the vaginal microbiota has been well-studied.” (L359), and “Healthy reproductive tract microbiota mediate critical functions in the host, such as development of the immune system, protection against opportunistic infections, facilitation of digestion and production of bioactive metabolites.” (L360), but it is difficult for readers to judge it without references as evidence for each description. Authors should add references for these descriptions.

Thank you for your comment. We added references.

Minor comments

L179: “RNAlater solution” may be “RNA later solution”.

Thank you for your comment, RNAlater is correct as written. It is the trade name.

L179: “at 4 ºC” may be “at 4 ºC”.

L185: “(IL-1 )” may be “(IL-1)”.

Thank you for your comment, the editor’s version did not show beta after IL-1.

L378: “the cardiovascular system (heart and aorta)” may be “vascular sites (heart and aorta)”.

Thank you for your comment, we corrected it.

L381: “vascular sites (heart, aorta)” may be “vascular sites (heart and aorta)”.

 Thank you for your comment, we corrected it.

Reviewer 4 Report

Comments and Suggestions for Authors

I found all components of the manuscript to be complete and well done. 

Author Response

Dear Reviewer:

Thank you so much for your kind comment. 

Best

Changchang Ye

Round 2

Reviewer 2 Report

Comments and Suggestions for Authors

Dear authors,

The revised manuscript is substantially improved and I have no further suggestions.

Reviewer 3 Report

Comments and Suggestions for Authors

Authors revised the manuscript, but, this article has not fully answered some of the questions due to inadequate statistical analysis and insufficient description.

First, authors suggest “The figure and table were corrected.”, but figure 2 is reversed, and table 1 is too small. Moreover, the number of samples is not shown in table 1. Authors should revise the manuscript, carefully.

Second, authors used range in figure 2, figure 3, figure 4 figure 7 and figure 8, but they do not explain what they were in figure legend. Authors should revise the manuscript, carefully.

Third, authors suggest “Similarly, T. denticola showed a trend (not significant) toward higher levels in the infection group” (L282), but the sample size is only 6, it is difficult to conclude the association as “trend”. Authors should rewrite the results based on results of the statistical analysis.

Fourth, as mentioned in the first review, authors suggest “P. gingivalis colonization in the uterus was significantly increased after the polymicrobial periodontal infection.” (L301), but they do not show the results of statistical analysis (i.e., the difference between “before” and “after”). In fact, authors suggest “we do not have before and after data, We have INF group compared to control group”. If so, authors could not use the word “increased”. Furthermore, as mentioned in the first review, authors suggest “Although nisin trended to decrease the levels of P. gingivalis in the 304 ovary and T. forsythia in the uterus and vagina, these changes were not significant.” (L301), “nisin treatment increased both the Shannon and Simpson index in the uterus” (L335), “”only the change in the Shannon index reached statistical significance.” (L336), “s. Although nisin treatment compared to infection significantly elevated the Chao index in the ovary and heart.” (L340), “Nisin treatment alone or in the context of infection increased the observed species in the uterus, ovary, vagina, and heart.” (L341), “nisin treatment alone or in the context of infection increased the Pseudomonas and decreased the Muribaculaceae, and Lachnospiraceae NK4A136 groups in the reproductive organs and heart tissue (Figure 6b, 8).” (L393) “Moreover, the polymicrobial periodontal infection increased the Turicibacter group in the reproductive organs and heart (Figure 8).” (L396), “The current study also showed an elevated host immune cytokine (TNF- and IL-6) response in the oral cavity and uterus as a result of the polymicrobial infection, indicating a local and systemic inflammatory response to the microbes and microbial products.” (L459), but they do not show the levels before the interventions, and therefore it is difficult to conclude the change of levels (e.g., “decrease”, “increased”, “change” and “elevated”). In fact, authors suggest “We compared to CTL group.”, but if so, authors could not use the words such as “increased”, “increased”, “change”, and “elevated”. Authors should describe the results sections based on the results.

Fifth, as mentioned in the first review, authors showed the results in aorta in figure 2, but they do not show them in figures. Authors suggest “the DNA from aorta is limited for subsequent 16s RNA sequencing analysis. The aorta tissue was very limited. The quantity of aorta DNA was not enough after PCR and cytokine examinations.”, but if so, it may limitation of this study. Authors should explain these reasons.

Finally, as mentioned in the first review, authors do not justify why the polymicrobial inoculum was not administered for every day, although they cited references. Authors should add descriptions for the justification.